# Nonmetallic Active Sites on Nickel Phosphide in Oxygen Evolution Reaction

**DOI:** 10.3390/nano12071130

**Published:** 2022-03-29

**Authors:** Pengfei Zhang, Hongmei Qiu, Huicong Li, Jiangang He, Yingying Xu, Rongming Wang

**Affiliations:** Beijing Advanced Innovation Center for Materials Genome Engineering, Beijing Key Laboratory for Magneto-Photoelectrical Composite and Interface Science, School of Mathematics and Physics, University of Science and Technology Beijing, Beijing 100083, China; zpf3516@163.com (P.Z.); hmqiu@ustb.edu.cn (H.Q.); 18166758766@163.com (H.L.); jghe2021@ustb.edu.cn (J.H.)

**Keywords:** Ni_12_P_5_, oxygen evolution reaction, adsorption energy, nonmetallic active site, ab initio molecular dynamics, density functional theory

## Abstract

Efficient and durable catalysts are crucial for the oxygen evolution reaction (OER). The discovery of the high OER catalytic activity in Ni_12_P_5_ has attracted a great deal of attention recently. Herein, the microscopic mechanism of OER on the surface of Ni_12_P_5_ is studied using density functional theory calculations (DFT) and ab initio molecular dynamics simulation (AIMD). Our results demonstrate that the H_2_O molecule is preferentially adsorbed on the P atom instead of on the Ni atom, indicating that the nonmetallic P atom is the active site of the OER reaction. AIMD simulations show that the dissociation of H from the H_2_O molecule takes place in steps; the hydrogen bond changes from O^a^-H⋯O^b^ to O^a^⋯H-O^b^, then the hydrogen bond breaks and an H^+^ is dissociated. In the OER reaction on nickel phosphides, the rate-determining step is the formation of the OOH group and the overpotential of Ni_12_P_5_ is the lowest, thus showing enhanced catalytic activity over other nickel phosphides. Moreover, we found that the charge of Ni and P sites has a linear relationship with the adsorption energy of OH and O, which can be utilized to optimize the OER catalyst.

## 1. Introduction

The increasing global energy demand and environmental pollution make it imperative to develop more efficient and sustainable energy conversion technologies [1,2,3,4]. Hydrogen is one of the most plentiful elements in the universe and the most efficient green fuel. However, most hydrogen on earth only exists in water molecules. Electrocatalytic water splitting is one of the most promising technologies for hydrogen generation [5,6,7]. At present, the efficiency of electrocatalytic water splitting is too low to meet the requirements of large-scale applications. Electrocatalytic water splitting consists of two half-reactions, the oxygen evolution reaction (OER) and the hydrogen evolution reaction (HER) [8]. The OER involves the transfer of four electrons, which usually has a relatively high overpotential and constrains the efficiency of water splitting; therefore, more efficient catalysts for OER have to be carefully designed in order to accelerate the reaction [9,10].

On the other hand, at present the most efficient electrocatalysts for OER reactions are Ir/Ru/Pt-based noble metals or their oxides [11,12,13]. However, the high cost of these noble metal-containing electrocatalysts significantly impedes their large-scale application in industrial production. Therefore, the development of high-performance and low-cost catalysts is highly desirable. Recently, cheap transition metal oxides [14,15], phosphides [16,17,18,19], nitrides [20], sulfides [21,22,23,24], carbides [25], etc., have been studied as emerging electrocatalysts, and many show excellent OER catalytic activity.

Among these, nickel phosphides (Ni_2_P, Ni_3_P, Ni_5_P_4_, Ni_12_P_5_, etc.) have attracted a great deal of attention recently [26,27,28,29,30,31]. In particular, high OER catalytic activity has been reported in Ni_12_P_5_. Menezes demonstrated that Ni_12_P_5_ shows stronger OER performance compared to Ni_2_P [31]. Xu et al. found that Au/Ni_12_P_5_ core/shell nanoparticles (NPs) show high OER catalytic activity and that a synergetic effect exists in the single crystalline core/shell structure [32]. The OER activity of Ni_2_P has been extensively studied theoretically [33,34,35]. However, there are few theoretical studies on the OER mechanism of Ni_12_P_5_, especially studies combining theory with molecular dynamics. In a previous study, Wen et al. verified that the rate-determining step for the OER of Ni_12_P_5_ is the formation of the OOH group, and the energy barrier is 1.58 eV; however, the OER active site of Ni_12_P_5_ has not been thoroughly studied [36].

Density functional theory (DFT) has been widely used to analyze the electrochemical water splitting process and study the catalytic activity of OER [37,38,39,40]. Ab initio molecular dynamics (AIMD) provides microscopic insights into the structural and dynamical properties of aqueous solutions, which serves as a perfect supplement to experimental studies [41,42,43,44]. In order to provide a basis for understanding the catalytic mechanism of Ni_12_P_5_, in this work we used AIMD and DFT calculations to study the OER catalytic process on nickel phosphides. It was found that the dissociative adsorption of H_2_O is at the nonmetallic P site, not the Ni site. In addition, we found that the adsorption energies of OH and O are linearly correlated with the amount of charge at the adsorption sites. Furthermore, our calculations show that the overpotential of Ni_12_P_5_ is lower than in other nickel phosphides, which explains its excellent OER catalytic activity. This study unveils the microscopic mechanism of OER on the surface of Ni_12_P_5_ and provides a general picture of charge transfer on nickel phosphides, which can benefit catalyst design in the future.

## 2. Computational Details

### 2.1. Static DFT Calculations

In the static calculations, the spin-polarized DFT calculations were performed using the Vienna Ab initio Simulation Package (VASP) [45,46]. The generalized gradient approximation (GGA) of the Perdew–Burke–Ernzerhof (PBE) functional was applied to optimize the geometric structures [47]. The interactions between the ions and valence electrons were described using the Projector Augmented Wave (PAW) method [48]. The slab structure was relaxed, with fixed in-plane lattice constants along with the bottom layer until the force on each atom was less than 0.01 eV/Å. The convergence criterion of the total energy for all the calculations was set as 1 × 10^−5^ eV, and the *k*-point sampling within the Brillouin zone for different structures is shown in Appendix A. The plane wave basis was adopted for all of the calculations, with a cutoff energy of 400 eV.

The adsorption energy was calculated using the following equation:(1)$$E_{\mathrm{ads}}\left(X\right)=E_{*+X}-E_{*}-E_{X}$$
where $E_{*+X}$ is the DFT total energy of the slab with X species adsorbed on the surface and $E_{X}$ and $E_{*}$ are the total energies of X species and the slab with a clean surface, respectively.

An OER consists of four basic reaction steps, with each step involving one electron transfer and one proton removal [39]:$$\begin{array}{c} {}*+{\;H}_{2}O\to *\mathrm{OH}+H^{+}+e^{-}\\ {}*\mathrm{OH}\to *O+H^{+}+e^{-}\\ {}*O+H_{2}O\to *\mathrm{OOH}+H^{+}+e^{-}\\ {}*\mathrm{OOH}\to O_{2}+H^{+}+e^{-} \end{array}$$
in which * denotes a surface site and *X represents an adsorbed X intermediate on the surface. The reaction free energy for each step is calculated using
(2)$$\Delta G=\Delta E+\Delta E_{\mathrm{ZPE}}-T\Delta S$$
where ΔE, $\Delta E_{\mathrm{ZPE}}$, and ΔS are the ground state energies calculated by DFT, zero-point energy, and the entropy correction values, respectively [39,49,50]. Additional computational details are described in the Supporting Information.

### 2.2. AIMD Simulations

AIMD simulations were carried out using the CP2K package with the canonical NVT ensemble, employing Nosé–Hoover thermostats with a target temperature of 300 K [51]. The MD time step was set to 1.0 fs. The DFT in CP2K/Quickstep is based on the mixed Gaussian Plane Wave (GPW) [52]. The matrix diagonalization method was used to optimize the wave function. The double-ζ Gaussian basis set with one set of polarization functions (DZVP) was used [53], and the plane wave basis was set with an energy cutoff of 350 Ry. The O 2s2p, P 3s3p, and Ni 3s3p3d4s were treated as valence electrons and the other electrons were placed in the cores utilizing Goedecker–Teter–Hutter (GTH) pseudopotentials [54,55]. The Perdew–Burke-Ernzerhof (PBE) [56] exchange-correlation functional and the Grimme D3 method [57] of correction to the van der Waals interaction were adopted. 

The surface of Ni_12_P_5_ (001) was simulated by a *p* (2 × 2) supercell with four atomic layers. The vacuum space between the slab and its periodic image was 15 Å; therefore, the size of the Ni_12_P_5_ (001) supercell was 17.2 × 17.2 × 22.9 Å^3^. Ni_12_P_5_ (001)/H_2_O was simulated using the explicit solvent model. Water solution was added above the surfaces of Ni_12_P_5_ (001) with thickness of 11 Å and volumetric density of 1 g/mL. The Ni_12_P_5_ (001)/H_2_O model contains 144 Ni atoms, 60 P atoms, and 100 H_2_O molecules.

## 3. Results and Discussion

### 3.1. Stability of Ni_12_P_5_ Surface

Ni_12_P_5_ crystalizes in a body-centered tetragonal structure with a space group of I4/m, as shown in Figure 1a. The P atom has two Wyckoff positions in Ni_12_P_5_; one is located at the center of a cube formed by the 8-Ni-atom, and the other is located in the center of the polyhedron with the 10-Ni-atom (Figure 1b). The fully relaxed lattice constants of bulk Ni_12_P_5_ are a = b = 8.644 Å and c = 5.051 Å, which are within 0.3% of the error when compared with the experimental data (a = b = 8.646 Å and c = 5.070 Å, JCPDS 89-3697).

To determine the most stable surface of N_12_P_5_ for catalysis study, we calculated the surface energies of several different low-index surfaces using DFT (Appendix A). According to Reuter and Scheffler’s surface energy calculation method [58], the surface energy of N_12_P_5_ is defined as
$$E_{\mathrm{surf}}=\left[\frac{1}{2A}E_{\mathrm{slab}}-n_{\mathrm{Ni}}E_{\mathrm{Ni}\;\mathrm{bulk}}-n_{P}E_{P\;\mathrm{bulk}}-\frac{E_{f,{\mathrm{Ni}}_{12}P_{5}}}{12}+\Delta \mu_{P}\left(\frac{5n_{\mathrm{Ni}}}{12}-n_{P}\right)\right]$$
where $E_{\mathrm{slab}}$ is the total energy of the surface model, $E_{\mathrm{Ni}\;\mathrm{bulk}}$ and $E_{P\;\mathrm{bulk}}$ are the total energies per atom of metal Ni and black P, $E_{f,{\mathrm{Ni}}_{12}P_{5}}$ is the formation energy of Ni_12_P_5_, $n_{\mathrm{Ni}}$ and $n_{P}$ are the numbers of Ni and P atoms in the surface model, respectively, $\Delta \mu_{P}$ is the chemical potential of P, and *A* is the surface area of the surface model.

Five low-index surfaces of Ni_12_P_5_, namely (001), (100), (110), (101), and (111), are considered in this work; the calculated results are shown in Figure 1c. Within the thermodynamically stable district (−1.2 eV < $\Delta \mu_{P}$ < −0.4 eV) of Ni_12_P_5_ [59], the surface energies of these surfaces follow the order (001) < (100) < (110) < (101) < (111). Surface (001) has the lowest $E_{\mathrm{surf}}$, and hence is the most stable surface. Therefore, we focus on surface (001) in this work.

### 3.2. The Adsorption and Dissociation of H_2_O on the Surface of Ni_12_P_5_

AIMD simulations with an explicit solvent model (Figure 1d) were performed for several Ni_12_P_5_ (001)/H_2_O models, with random distribution of initial water molecules in NVT ensemble at 300 K. Movies of the reaction trajectories are provided in the Supporting Information (Appendix A). The explicit solvent model can capture the interactions among H_2_O molecules in aqueous solution, and is therefore able to describe the dynamic process of H_2_O molecules at the surface more accurately.

Individual snapshots from the MD trajectory of H_2_O at the Ni_12_P_5_ (001)/H_2_O are displayed in Figure 2a. During the entire simulation process, an H_2_O molecule is first adsorbed on the top of a P atom on the surface of Ni_12_P_5_ (001) and then an H^+^ is dissociated into the solution, which corresponds to the reaction of *OH_2_→*OH+H^+^+e^−^ The H_2_O dissociation process can be studied in detail by monitoring the distances between the involved atoms as a function of the simulation time. As shown in Figure 2b, we can see that the whole process can be divided into the following periods (where O^a^ is the O atom in the adsorbed H_2_O and *OH and O^b^ is the O atom in another H_2_O in solution):(a)Period I: 0–320 fs; H_2_O approaches the surface of Ni_12_P_5_ (001).

At first, due to thermal motion at room temperature, an H_2_O molecule in the aqueous solution moves towards the Ni_12_P_5_ (001) surface, which corresponds to the decreasing P-O^a^ distance. While the P-O^a^ distance approaches ~2.00 Å, the H-O^a^ bond length of the H_2_O molecule maintains ~0.98 Å, which corresponds to the H-O bond length of H_2_O gas. Meanwhile, a hydrogen bond, O^a^-H⋯O^b^ (2.65 Å, ⋯ denotes a hydrogen bond), is formed. 

(b)Period II: 320–375 fs. H_2_O is adsorbed and H dissociation starts; the hydrogen bond transforms.

With the P-O^a^ distance decreasing, one H^+^ of the H_2_O^a^ molecules becomes more and more active and begins to dissociate from the H_2_O^a^ molecule. At 375 fs, the P-O^a^ distance decreases to 1.65 Å, which is close to the P-O σ bond length (1.61 Å) in P_2_O_5_, indicating that the P-O^a^ bond is formed. Meanwhile, one of the H-O^a^ bond lengths increases from 0.98 Å to 1.60 Å, and the dissociated H^+^ forms H_3_O^+^ with another H_2_O molecule. In this process, the hydrogen bond transforms from P-O^a^-H⋯O^b^ to P-O^a^⋯H-O^b^; the hydrogen bonds length are 2.65 Å.

(c)Period III: 375–460 fs; the metastable state, while P-O^a^ is connected with an H through the hydrogen bond, P-O^a^⋯H-O^b^.

The O^a^ atom remains bonded with the P atom, and the P-O^a^ bond length is shortened to 1.65 Å. The hydrogen bond P-O^a^⋯H-O^b^ remains stable. 

(d)Period IV: After 460 fs; the breaking of the hydrogen-bond, H^+^ completely leaves the surface.

The hydrogen bond P-O^a^⋯H-O^b^ starts to break, and H_3_O^b+^ leaves the surface site. 

The similar processes of H_2_O dissociative adsorption from the two other AIMD simulations with different initial water molecules are shown in Appendix A. There are two rapid processes: First, the process of P-O^a^ bond shortening from 2.00 Å to 1.65 Å in Period II takes around 50~75 fs, which is the H_2_O chemical adsorption on Ni_12_P_5_ (001) surface. Second, the dissociation of the hydrogen bond from the H_2_O molecule (the O^a^-H bond length increases from 1.60 Å to over 2.00 Å) in Period IV takes ~50 fs experimentally, Z.-H. Loh et al. found the time to form OH and H_3_O^+^ from the ionized H_2_O^+^ and H_2_O is about 46 fs [60], which is consistent with what we found from AIMD simulation. Because the energy required to break O-H bond is much larger than to break a hydrogen bond, the step-by-step dissociation of H^+^ with catalyst is thermodynamically more favorable. 

On the other side, the metastable state with partially dissociated H and hydrogen-bond P-O^a^⋯H-O^b^ in Period III can persist as long as 1 ps (Appendix A). A hydrogen bond widely exists between two H_2_O molecules (about ~0.21 eV/bond in H_2_O [61]) in solutions and is the main intermolecular interaction in the liquid [62,63]. In the Period III, the state with hydrogen bond structure may be metastable.

We further verified these results with radial distribution functions (RDF). As can be seen from the RDF plot of Ni and P atoms in Figure 2c, the distances of Ni-P do not change significantly, which indicates that the Ni_12_P_5_ (001) surface is stable in aqueous solution. Figure 2d shows the RDF between O atoms in H_2_O and one P atom at the Ni_12_P_5_ (001) surface. At 0 fs the P-O distance is around 4 Å, which corresponds to the start point, when H_2_O molecules are not adsorbed on the surface yet. At 200 fs, a peak appears at 2.45 Å, indicating that H_2_O is moving towards the Ni_12_P_5_ surface. At 400 fs, the peak shifts to 1.65 Å. From 400 fs to 1800 fs, the peak keeps its shape and position at 1.65 Å, indicating a stable P-O bond has been formed and the H_2_O is adsorbed on the surface of Ni_12_P_5_ (001). Figure 2e shows the RDF of the O and Ni atoms on the surficial layer. We found no any peak in the RDF within the range of r < 3 Å, which means no H_2_O molecules were adsorbed to Ni atoms. After 1000 fs there is a small peak appearing in the range of 2~2.5 Å; it can be inferred that after *OH is bonded to the P site, the H_2_O molecule in solution may approach the Ni site during thermal motion. The time evolution of the Ni-O distance is detailed in Appendix A. No stable Ni-O bond was found in any of our simulations. 

From the above discussion, the adsorption and dissociation process of H_2_O on the surface of Ni_12_P_5_ is captured in the AIMD simulations. In this process, the transformation and fracture of the hydrogen bond plays an important role. It should be noted that the H_2_O molecule prefers to adsorb on the nonmetal P site instead of the metal Ni site, which is uncommon in transition metal compounds [37,38,39], and will be discussed in the next section.

### 3.3. Nonmetallic P Atoms as Active Sites

Transition metals with partially filled d orbitals are generally the active catalytic sites in OER [16,17,21,64]. Recently, it has been reported that nonmetal atoms may become the active sites for catalytic reactions in theoretical studies [63,64,65]. For example, Legare et al. found that B atoms could be active sites in nitrogen reduction [65]. Deng et al. pointed out that in Pt-loaded MoS_2_ the active site are the surface S atoms directly connected to Pt [66]. Huang et al. showed that in Ni-N_4_-Cs, OH and O tend to adsorb on the second adjacent C atom, while OOH and OO are formed on the central Ni atom [67]. It was pointed that the interaction between the central metal atom and its adjacent coordination atoms can tune the catalytic performance from the aspects of electronic structure, spatial coordination, etc. Thus, the coordinated nonmetal atoms may act as the active sites for the catalytic processes [68].

The stabilities of adsorption groups at the P site on Ni_12_P_5_ surface were further examined. An adsorbed group (OH or O) was set on the top of the P and Ni sites, respectively, of the Ni_12_P_5_ (001) surface and the relative energies of these two configurations were computed by using more accurate static DFT calculations. As can be seen from Figure 3, the total energy of the system decreased from Ni*OH (*O) to P*OH (*O), and the energy difference (ΔE) between these two configurations was −1.15 eV (−2.56 eV), which indicates that OH (O) prefers to adsorb on the P atom rather than the Ni of the Ni_12_P_5_ (001) surface. In addition, our AIMD simulations (Appendix A) show that the adsorbed OH on the P atom remains stable, while OH adsorbed on Ni moves to its adjacent P atom in 450 fs (Figure 3a). Similarly, the adsorbed O on Ni moves to the nearby P atom in 150 fs (Figure 3b), and the adsorbed O on P atom remains stable.

As discussed above, we found that the OER active site on the surface of Ni_12_P_5_ is the nonmetal P site; we discuss the active sites and catalytic activity of OER in several nickel phosphides in Section 3.4. In the catalytic process, the electronic structure of the nonmetallic P active sites may be affected by coordinated Ni atoms, which is detailed in Section 3.5.

### 3.4. Active Site of OER in Nickel Phosphides

In this subsection, we investigate the electronic properties of active sites and the catalytic activities of several nickel phosphides using the static DFT method. The surface electrostatic potential of Ni_2_P, Ni_12_P_5_, Ni_5_P_2_, Ni_3_P, and NiO are shown in Figure 4a–e. The typical OER electrocatalyst NiO, the active site of which is Ni, is used as a reference. The Ni-O bond on NiO is mainly ionic and the electrostatic potential varies sharply. Compared with NiO, the electrostatic potential fluctuations of the Ni_2_P, Ni_12_P_5_, Ni_5_P_2_ and Ni_3_P surfaces are smaller. Due to the small electronegativity difference between P and Ni, P is only slightly negative charged, while Ni is slightly positively charged; therefore, the Ni-P bond is mainly covalent.

Different surface structures that satisfy the nickel and phosphorus stoichiometric ratio of the compound were selected in order to calculate the adsorption energies of OH and O (Appendix A). As shown in Figure 4f, the adsorption energies of OH (O) at the P site are always lower than that of Ni on the Ni_2_P, Ni_12_P_5_, Ni_5_P_2_ and Ni_3_P surfaces, indicating that the active catalytic sites on these nickel phosphides are P atoms. Moreover, the adsorption energies of OH and O on the surface of Ni_12_P_5_ are the lowest among all the studied nickel phosphides in this work. We noted that P has more suspended bonds than Ni atoms on the exposed surfaces of Ni_2_P, Ni_12_P_5_, Ni_5_P_2_, and Ni_3_P (Appendix A). In general, step, apex, and highly unsaturated coordination atoms usually tend to be active sites of OER [69,70], which may explain why H_2_O tends to adsorb on P site.

Previous studies have shown that large charge transfers between active sites and adsorption groups usually correspond to lower adsorption energies. To unveil the underlying mechanism of the adsorption behavior of OH and O on nickel phosphides, we calculated the amount of charge transfer at the sites on the nickel phosphide surface when OH (or O) was adsorbed. As can be seen from Table 1, the electrons in P and Ni transfer to the adsorbed O atom, which is more electronegative. We found that the transferred charge is strongly correlated with adsorption energy, i.e., larger charge transfers between OH or O and adsorption sites are correlated with lower adsorption energy in the corresponding adsorption structure. The larger transferred charge indicates low charge transfer resistance between the adsorption site and the OH and O groups, which can promote the adsorption of these groups, thus showing excellent performance [71,72,73].

Because the adsorption energies of OH and O are related to the amount of transferred charge, there may be a correlation between the adsorption energy and the amount of charge at the adsorption site. Therefore, we calculated the charge of the Ni and P sites of the clear nickel phosphide surface (Figure 4g). It was found that the amount of charge at the active site has a linear relationship with the adsorption energy. As shown in Table 2, when the charge of the P atom is in the range of −0.4 to −0.15 |e|, the adsorption energy of OH or O decreases with the increase of charge. When the charge of Ni is in the range of 0.05 to 0.2 |e|, the adsorption energy of OH or O is directly proportional to the charge.

As shown in Figure 5a, the Gibbs energies corresponding to each reaction step show an upward trend under standard conditions (U = 0 V), meaning that extra energy is needed to promote the reaction. Figure 5b shows that the electronic transfer step from *O to *OOH has the highest free energy gradient, indicating that it is the rate-determining step (RDS) for the OER process. Obviously, the third step energy barrier for Ni_12_P_5_ is 0.71 eV, which is lower than for the Ni_2_P (1.32 eV), Ni_5_P_2_ (2.15 eV), and Ni_3_P (2.49 eV) structures, indicating that Ni_12_P_5_ has best OER catalytic activity; this is consistent with the experimental results (Appendix A).

### 3.5. Charge Distribution at Ni_12_P_5_ (001)/H_2_O

In the OER catalytic process, the electronic structure of the nonmetallic P active sites may be affected by coordinated Ni atoms, which is discussed further in this section. Understanding the nature and magnitude of charge transfer of Ni_12_P_5_/H_2_O is helpful in understanding its synergistic action between different atoms. When the H_2_O molecule is absorbed to the P atom, the charge state of the molecule changes. From the charge density difference diagram (Figure 6a,b), we can see that there is strong charge accumulation/depletion between the P and O atoms. Because of the high electronegativity, O atoms tend to obtain electrons from adjacent P atoms. Bader charge analysis (Table 3) shows that the charge of a P atom bound to an O atom changes from −0.18 |e| to 0.87 |e|, among which −0.53 |e| transfer to the O atom and the other electrons transfer to the Ni atoms of Ni_12_P_5_, which causes the charge of the O atom to decrease from −1.11 |e| to −1.67 |e| (Figure 6c).

Li et al. found that the high oxidation state of Ni active sites, which implies high electron affinity, is conducive to OH^−^ adsorption and the OER reaction [74]. When the electron in the P atom is transferred to the surrounding Ni atoms, the Ni atoms carry more charge, and thus have low electron affinity, which is not conducive to OH adsorption. On the other hand, the P atoms carries more positive charge, making it easier to adsorb OH. It has been reported that the synergistic effect between atoms can promote charge transfer and accelerate the catalytic reaction [67,75,76]. Therefore, the synergistic action of P and Ni atoms can promote the charge transfer between atoms, and makes P atoms the active sites in OER.

## 4. Conclusions

In summary, we studied the structural evolution and the adsorption and dissociation of an H_2_O molecule on the surface of Ni_12_P_5_ in the aqueous condition using AIMD. The results showed that the H_2_O molecule is preferentially adsorbed on the P atom, indicating that the nonmetallic P atom is the active site of the OER reaction. Our simulations show that the adsorbed H_2_O molecule first forms a hydrogen bond with other H_2_O molecules, then the hydrogen bond changes from O^a^-H⋯O^b^ to O^a^⋯H-O^b^, and finally the hydrogen bond breaks and a H^+^ is dissociated. In the OER reaction, the RDS is the formation of the OOH group, and the overpotential of Ni_12_P_5_ is the lowest, thus showing better catalytic activity, which explains the experimental observations. Moreover, we found that the charge of Ni and P sites has a linear relationship with the adsorption energy of OH and O, which can be utilized to optimize OER catalysts. This work provides insights into the OER catalytic performance of nickel phosphide and the design strategy of achieving high OER activity in nickel phosphides.

## Figures and Tables

**Figure 1 nanomaterials-12-01130-f001:**
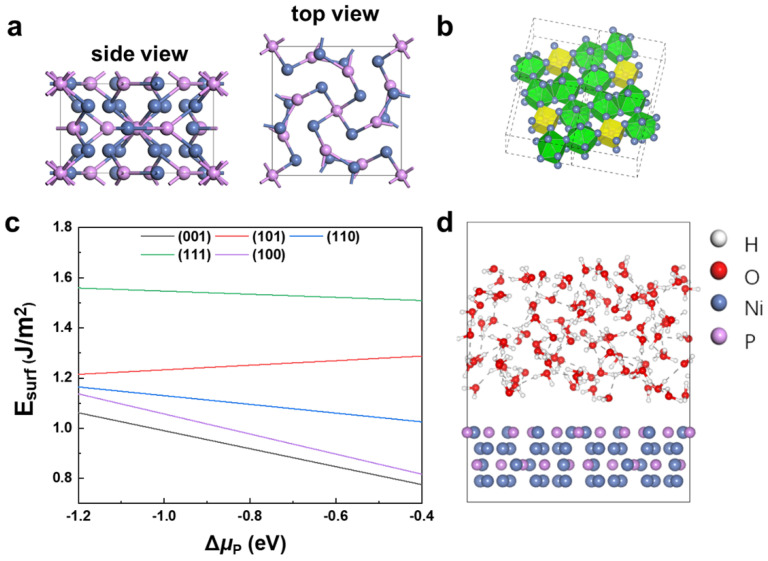
(**a**) Crystal structure of Ni_12_P_5_. (**b**) Equivalent sites of P ions. Yellow is the P-centered cube with 8-Ni-atom and green is P-centered polyhedron with 10-Ni-atom. (**c**) Surface energies of low-index surfaces (001), (101), (110), (100), and (111) as a function of P chemical potential (Δ$\mu_{P}$). (**d**) Ni_12_P_5_ (001)/H_2_O model. The white, red, purple, and blue spheres represent H, O, P, and Ni, respectively.

**Figure 2 nanomaterials-12-01130-f002:**
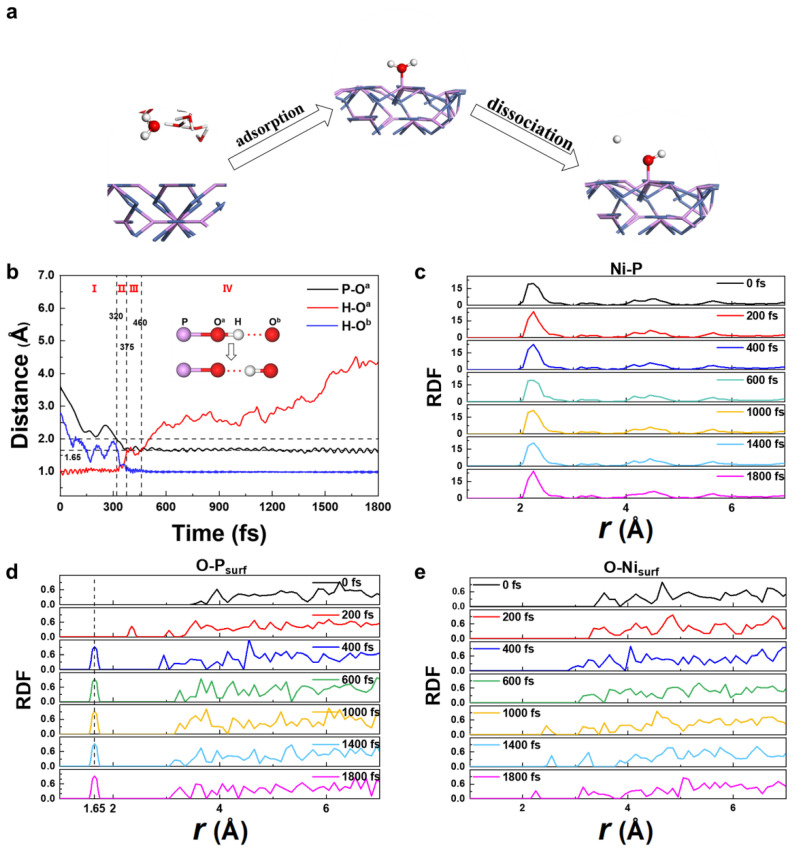
(**a**) Selected snapshots of the AIMD trajectory for Ni_12_P_5_ (001)/H_2_O at 300 K. The white, red, purple, and blue spheres represent H, O, P, and Ni, respectively. (**b**) The time evolution of the key bond distances during the AIMD simulation. The black line is the bond distance between the P-O^a^. The red line is the bond distance between the H-O^a^. The blue line is the bond distance between the H-O^b^. H is the dissociated H^+^, O^a^ is the O atom in *OH; O^b^ is the O atom in another H_2_O. Radial distribution function (RDF) changes over time: (**c**) among Ni and P atom; (**d**) P atoms on the surface and O atom in H_2_O; (**e**) Ni atoms on the surface and the O atom in H_2_O.

**Figure 3 nanomaterials-12-01130-f003:**
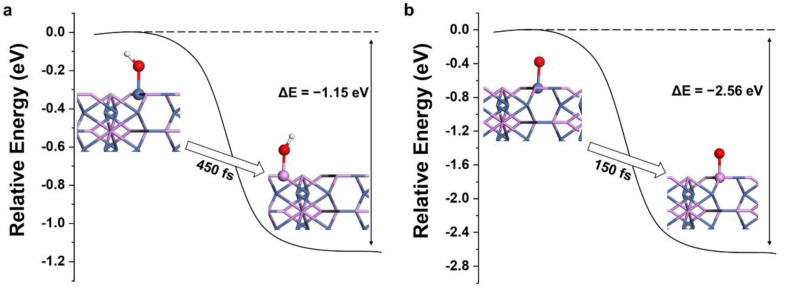
Relative energy distributions for different products on Ni_12_P_5_ (001) surface: (**a**) Ni*OH→P*OH (**b**) Ni*O→P*O. The white, red, purple, and blue spheres represent H, O, P, and Ni, respectively.

**Figure 4 nanomaterials-12-01130-f004:**
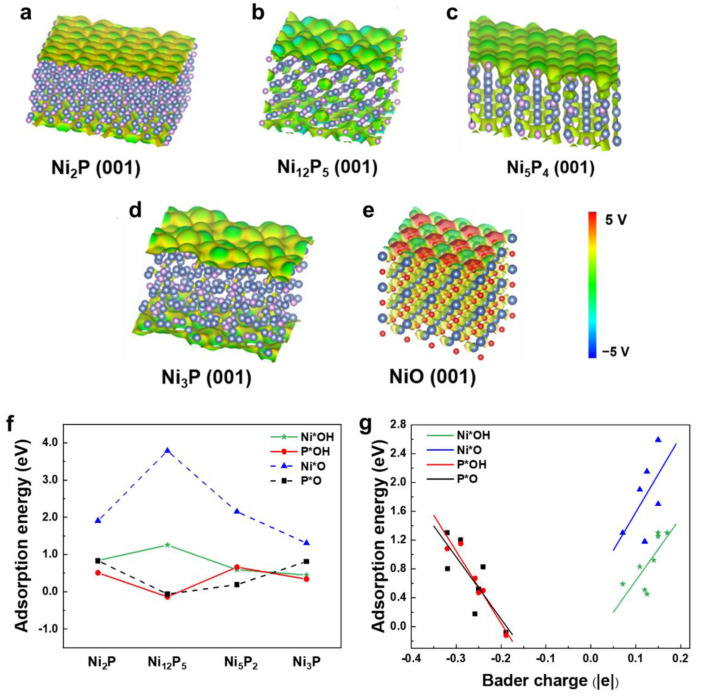
Surface electrostatic potential: (**a**) Ni_2_P (001), (**b**) Ni_12_P_5_ (001), (**c**) Ni_5_P_2_ (001), (**d**) Ni_3_P (001), and (**e**) NiO (001). The isosurface value is set to 0.2 e/Å^3^. (**f**) Adsorption energies of OH and O adsorbents on the surfaces of Ni_2_P (001), Ni_12_P_5_ (001), Ni_5_P_2_ (001), and Ni_3_P (001). (**g**) The relationship between adsorption energy (OH and O) and Bader charge of the atom (Ni or P) bonded with adsorbents.

**Figure 5 nanomaterials-12-01130-f005:**
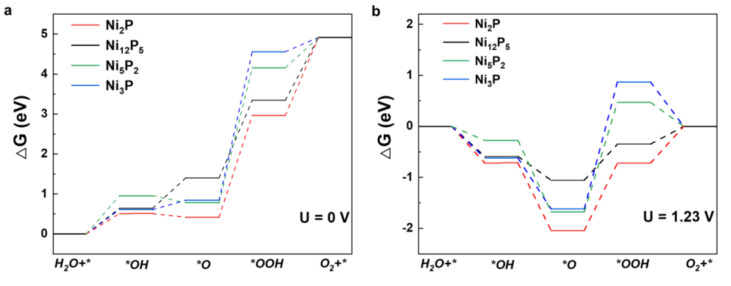
Gibbs energy diagram of OER intermediates on the catalyst surfaces at electrode potentials (**a**) U = 0 V and (**b**) U = 1.23 V.

**Figure 6 nanomaterials-12-01130-f006:**
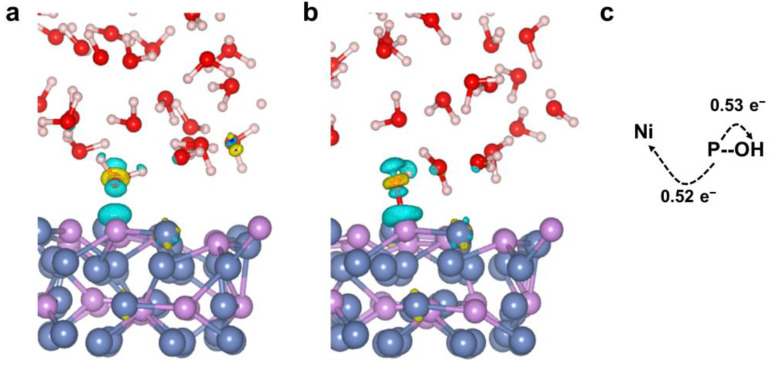
Electron density difference graphs: (**a**) P*OH_2_ and (**b**) P*OH. The accumulation of electron density is indicated by the yellow color, while the depletion of electron density is indicated by the blue color. (**c**) Direction of electron transfer. The white, red, purple, and blue spheres represent H, O, P, and Ni, respectively.

**Table 1 nanomaterials-12-01130-t001:** The amount of charge transfer of OH or O after adsorption, along with adsorption energies for nickel phosphide.

	Charge/e^−^	E_*ads*_/eV		Charge/e^−^	E_*ads*_/eV
*OH	0.37	2.54	*O	0.15	1.24
	0.85	1.71		0.21	0.92
	0.89	1.42		0.21	0.82
	0.91	1.31		0.40	0.69
	0.94	0.51		0.43	0.50
	0.95	0.53		0.44	0.45

(* denotes a surface site)

**Table 2 nanomaterials-12-01130-t002:** Formulas corresponding to different structures.

Structures	Formulas
Ni*OH	$E_{\mathrm{ads}}=8.72q-0.24$
Ni*O	$E_{\mathrm{ads}}=10.55q+0.53$
P*OH	$E_{\mathrm{ads}}=-9.97q-1.95$
P*O	$E_{\mathrm{ads}}=-8.58q-1.61$

(* denotes a surface site).

**Table 3 nanomaterials-12-01130-t003:** Bader Charge Analysis of *+H_2_O, *OH_2_ and *OH + H.

		H_2_O			Ni				
	P	O	H^1^	H^2^	Ni^1^	Ni^2^	Ni^3^	Ni^4^	Ni^5^
***+H_2_O**	−0.18	−1.11	0.55	0.56	0.21	0.21	0.11	0.12	0.08
***OH_2_**	0.66	−1.56	0.65	0.60	0.17	0.11	0.12	0.10	0.08
***OH + H**	0.87	−1.67	0.61	0.59	0.07	0.29	0.06	0.06	0.05

The data listed here are the Bader charges (in the unit |e|) of the atoms on the surface; P is the adsorption site of H_2_O on the surface, H^1^ is the dissociated H atom, H^2^ is the H atom in the interaction with *OH, Ni^1^, Ni^2^, and Ni^3^ are the nearest neighbors of P, and Ni^4^ and Ni^5^ are the next nearest neighbors of P.

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
