# Peer review of "Nonmetallic Active Sites on Nickel Phosphide in Oxygen Evolution Reaction"

_nanomaterials, 2022, doi:10.3390/nano12071130_

Round 1

Reviewer 1 Report

In this paper the author presented computational study of oxygen evolution reaction (OER) catalysed by nickel phosphide catalyst(s).  Utilizing ab initio molecular dynamics simulation and "static" DFT calculation it was shown that binding of H2O molecule is preffered on phosphorus rather than on nickel atom. Further, a profile of four step reaction profile of OER mechanism is calculated. The applied methodology in my opinion is well chosen and the authors presented results clearly. The authors cited most of the relevant publications. 

However, I have some questions and comments which could be addressed before publishing the paper. On Figure 5 Gibbs free-energies (I would rather call it Gibbs energy in accordance with IUPAC recommendation) of adsorption are presented and the energies seems quite high. It would be good to make a connection with some experimental findings and/or some other computational study of a similar processes. Further, if the figure represents the "reaction coordinate "it should include transition states that connects the minima (though they are just proton transfers. In conclusion the author stated that there is a liner relationship between atomic charges (Bader) and adsorption energy. However, there is no figure which presents this linear relationship. 

Reviewer 2 Report

The paper focuses on the microscopic mechanism of OER on Ni12P5 surface by using density functional theory calculations and ab initio molecular dynamics simulation. The authors demonstrate that water molecule is preferentially adsorbed on P atom, instead of the Ni atom, indicating the nonmetallic P atoms are the active sites of OER reaction.

The paper is generally well written and provide interesting results on the mechanism the structural evolution and the adsorption and dissociation of water molecule on Ni12P5 surface.

(a) the subject matter is suitable for the Journal;

(b) the quality of the presentation is appropriate;

(c) the work contains new and original contributions;

(d) appropriate reference to previous work is given;

(e) the conclusions are sound and justified;

(f) the abstract is informative;

(g) the title reflects the contents appropriately.

I suggest minor revisions according to the following point:

  • In the introduction section it is not clear how much the idea of the paper is novel. I suggest citing previous papers on density functional theory calcula- 13 tions (DFT) and ab initio molecular dynamics simulation (AIMD) applied to Ni12P5 catalyst evidencing the progress introduced by the paper.

Author Response

We thank reviwer for this suggestion. In the revised manuscript, we added sentences of “Theoretically, the OER activity of Ni2P has been extensively studied. However, there are few theoretical studies on the OER mechanism of Ni12P5, especially combining with molecular dynamics. In a previous study, Wen et al. verified that the rate-determining step for the OER of Ni12P5 is the formation of the OOH group, and the energy barrier is 1.58 eV, but the OER active site of Ni12P5 has not been thoroughly studied.”(Line 50-54 on Page 2). (Relevant reference: J Colloid Interface Sci 2020, 561, 638-646, ACS Catalysis 2019, 9, 8882-8892, Adv Mater 2019, 31,1901174 and J. Mater. Chem. A, 2021, 9, 9918–9926)